# BARcode DEmixing through Non-negative Spatial Regression (BarDensr)

**Shuonan Chen**[1,2,3,4,5,6]*, **Jackson Loper**[1,2,3,4,5,7], **Xiaoyin Chen**[8], **Alex Vaughan**[8], **Anthony M. Zador**[8], **Liam Paninski**[1,2,3,4,5,7]

**1** Mortimer B. Zuckerman Mind Brain Behavior Institute, Columbia University, New York, New York, United States of America, **2** Department of Statistics, Columbia University, New York, New York, United States of America, **3** Center for Theoretical Neuroscience, Columbia University, New York, New York, United States of America, **4** Grossman Center for the Statistics of Mind, Columbia University, New York, New York, United States of America, **5** Department of Neuroscience, Columbia University, New York, New York, United States of America, **6** Department of Systems Biology, Columbia University, New York, New York, United States of America, **7** Data Science Institute, Columbia University, New York, New York, United States of America, **8** Cold Spring Harbor Laboratory, Cold Spring Harbor, New York, United States of America

\* sc4417@cumc.columbia.edu

**Data Availability Statement:** The data underlying the results presented in the study is available from https://github.com/jacksonloper/bardensr.

**Funding:** This work was supported by the National Institutes of Health [NIH 5RO1NS073129,

## Abstract

Modern spatial transcriptomics methods can target thousands of different types of RNA transcripts in a single slice of tissue. Many biological applications demand a high spatial density of transcripts relative to the imaging resolution, leading to partial mixing of transcript rolonies in many voxels; unfortunately, current analysis methods do not perform robustly in this highly-mixed setting. Here we develop a new analysis approach, *BARcode DEmixing through Non-negative Spatial Regression* (BarDensr): we start with a generative model of the physical process that leads to the observed image data and then apply sparse convex optimization methods to estimate the underlying (demixed) rolony densities. We apply Bar-Densr to simulated and real data and find that it achieves state of the art signal recovery, particularly in densely-labeled regions or data with low spatial resolution. Finally, BarDensr is fast and parallelizable. We provide open-source code as well as an implementation for the 'NeuroCAAS' cloud platform.

## Author summary

Spatial transcriptomics technologies allow us to simultaneously detect multiple molecular targets in the context of intact tissues. These experiments yield images that answer two questions: which kinds of molecules are present, and where are they located in the tissue? In many experiments (e.g., mapping RNA expression in fine neuronal processes), it is desirable to increase the signal density relative to the imaging resolution. This may lead to mixing of signals from multiple RNA molecules into single imaging voxels; thus we need to *demix* the signals from these images. Here we introduce BarDensr, a new computational method to perform this demixing. The method is based on a forward model of the imaging process, followed by a convex optimization approach to approximately 'invert'

5RO1DA036913, RF1MH114132, U19MH114821, and U01MH109113 to A.M.Z., and 1U19NS107613 to L.P.], the Brain Research Foundation (BRF-SIA-2014-03 to A.M.Z.), IARPA MICrONS [D16PC0008 to A.M.Z. and D16PC0003 to L.P.], Paul Allen Distinguished Investigator Award [to A.M.Z.], Simons Foundation [350789 to X.C.], Chan Zuckerberg Initiative (2017-0530 ZADOR/ALLEN INST(SVCF) SUB awarded to A.M.Z and 2018-183188 to L.P.], and Robert Lourie (to A.M.Z.). This work was additionally supported by the Assistant Secretary of Defense for Health Affairs endorsed by the Department of Defense, 1120 Fort Detrick, Fort Detrick, MD 21702 through the FY18 PRMP Discovery Award Program W81XWH1910083 (to X.C). The funders had no role in study design, data collection and analysis, decision to publish, or preparation of the manuscript.

**Competing interests:** The authors have declared that no competing interests exist.

mixing induced during imaging. This new approach leads to significantly improved performance in demixing imaging data with dense expression and/or low spatial resolution.

This is a *PLOS Computational Biology* Methods paper.

## Introduction

Understanding the spatial context of gene expression in intact tissue can facilitate our understanding of cell identities and cellular interactions. How do neighboring cells' gene expressions relate to each other? How are different cell types with different activity patterns positioned in relation to each other? Is the subcellular distribution of gene expression informative about cell type or state? Multiplexed spatial transcriptomics methods, in particular *in situ* sequencing and sequential fluorescence *in situ* hybridization (FISH) methods [1, 2], offer a promising path to investigate these questions, allowing us to spatially resolve gene expression patterns at a single cell resolution. These assays can measure thousands of different genes simultaneously by looking at the same slice of tissue multiple times through multiple rounds of imaging. Using small barcoded sequences ('probes') which bind to target transcripts and amplify (generating easily detectable 'rolonies'), we can get exponentially more information about the nature of the tissue with each successive round of imaging.

However, fully exploiting this new data type can be challenging, for many reasons. Insufficient optical resolution can cause parts of multiple rolonies to appear in the same imaging voxel, resulting in a 'mixed' signal [3, 4]. Tissue can deform or drift over multiple rounds of imaging [5], and the signal from individual rolonies can vary slightly between imaging rounds [6]. The chemical washes may fail to complete their work in a given round, such that the imaging in the next round contains residual signal from the previous round (leading to a 'ghosting' effect). Some rolonies may entirely fail to bind to any probes in a given round [2, 3]. Most of these problems are rare, but they combine to yield a complex relationship between the signal of interest and the observed data.

Traditional techniques for extracting meaning from these images rely on good image pre-processing and clever heuristics; there are two main approaches that we are aware of. Both work well in ideal conditions. One school of thought ('blobs-first') begins by trying to identify regions in the tissue where a rolony may be present, and then tries to use the imaging data to guess the barcode identity of each rolony [4, 5, 7–9]. Another school of thought ('barcodes-first') begins by looking at each voxel and trying to determine whether the fluorescence signal emitted in that voxel over all the rounds is consistent with one of the barcodes [6, 10, 11]. These two approaches are implemented in e.g. the '*starfish*' (https://github.com/spacetx/starfish) package (under the names of 'spot-based' and 'pixel-based' approaches, respectively).

Both of these general approaches face difficulties whenever several rolonies make contributions to the same voxel. This situation arises for two reasons. First, it is desirable to maximize the signal density, to increase the number of transcripts detected per cell and therefore the power of any downstream statistical analyses. Second, it is desirable to minimize imaging time and file size by using lower imaging resolution. Both of these features—high density and insufficient optical resolution—lead to cases where different signals are mixed together into the same voxel. In this situation, to correctly identify rolony positions and identities it is necessary to perform some kind of 'demixing.' Because of this challenge, many current methods simply discard any blobs in regions where strong mixing occurs [3, 8, 9].

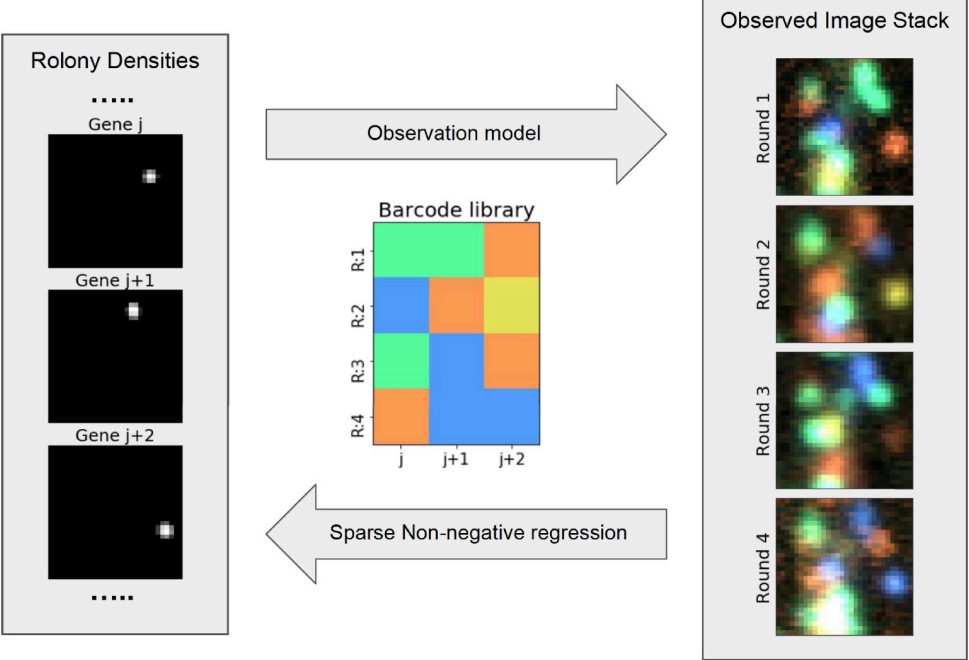

**Fig 1. BarDensr uses non-negative regression to demix and deconvolve the observed image stack, yielding a sparse intensity image for each barcode.** The key task of spatial transcriptomics data analysis is to take a stack of images (right) and use it to infer the locations of rolonies in the tissue. To solve this problem, BarDensr posits an 'observation model': a description of the physical process by which rolonies in the tissue give rise to the brightnesses we observe at each voxel. In particular, BarDensr assumes there is an unobserved 'rolony density' for each gene at each voxel (left), and the observation model mathematically describes how this rolony density transforms into the image stack we can see (right). Once this observation model is formulated, we can use sparse regression to solve the inverse problem: starting from an image stack, the regression gives us the value of the (unobserved) rolony density.

To overcome this challenge, we sought to address the multiplexing problem directly. *BARcode DEmixing through Non-negative Spatial Regression* (BarDensr) is a new approach for detecting and demixing rolonies. This approach directly models the physical process which gives rise to the observations (Fig 1), including background-noise components, color-mixing, the point-spread function of the optics, and several other features. By directly modeling these physical processes, we are able to accurately estimate overall transcript expression levels—even when the transcript density is so high that it is very difficult to isolate and decode the identity of individual rolonies.

BarDensr is designed to be incorporated in a full spatial transcriptomics imaging analysis pipeline. Note that BarDensr should not be applied directly to raw data from a microscope; registration and filtering appropriate to one's specific experimental data should be applied first. We give some examples in S1 Appendix (Section A) for appropriate preprocessing for the data used in this paper. We provide a Python package for implementing the BarDensr algorithm on either CPU or GPU architectures (https://github.com/jacksonloper/bardensr). The method requires about two minutes of compute time on a `p2.xlarge` Amazon GPU instance to process a seven-round, four-channel $1000 \times 1000$-voxel field of view from an experiment targeting 79 different transcripts. We also provide an implementation for the NeuroCAAS web-service [12], which can be used in a drag-and-drop fashion, with no installation required. We compared this method with three alternatives: the 'spot-based' and 'pixel-based' methods of *starfish*; a 'blobs-first' approach (Single Round Matching, or SRM, based on methods from [5, 8]); and a 'barcodes-first' approach (Correlation approach, or 'corr,' based on [6,

10, 11]). Both in simulation and real data, BarDensr improves on the state of the art in demixing accuracy, especially cases where denser and/or lower-resolution images are being analyzed.

## Methods

### Ethics statement

The Cold Spring Harbor Laboratory Animal Care and Use Committee approved all animal procedures, which were carried out in accordance with the Institutional Animal Care and Use Committee protocol 19-16-10-07-03-00-4 at Cold Spring Harbor Laboratory.

### Data

All the experimental data were obtained either using an improved version of BARseq [13] targeting 79 endogenous mRNAs in mouse visual cortex, or from a recent study [14] targeting 65 endogenous mRNAs in mouse motor cortex. Gene identities were read out using a seven-nucleotide gene identification index (GII), which was designed with a minimal Hamming distance of three nucleotides between each pair of GIIs. Rolonies were prepared as described by [14]. Imaging was performed on an Olympus IX81 inverted scope with a Crest Xlight2 spinning disk confocal, a Photometrics BSI Prime camera, and an 89 North LDI 7-line laser source. All images were acquired using an Olympus UPLFLN 40x 0.75 NA objective. The microscope was controlled by micro-manager [15]. See S1 Appendix (Section A) for the preprocessing steps for this data, and S1 Appendix (Section G) for the process of generating the simulation data.

### Notation and observation model

Formally speaking, what is the result of a spatial transcriptomics imaging experiment? For each voxel ($m$) in the tissue, at each imaging round ($r$), in each color-channel ($c$), we record a fluorescence intensity. We will use $\mathbf{X}_{m,r,c}$ to denote this fluorescence intensity. Our task is to use $\mathbf{X}$ to uncover the presence and identity of rolonies in the tissue. (Throughout we assume that $\mathbf{X}$ is preprocessed, including background removal and image registration, hence that there are no systematic shifts of the image between imaging rounds. See S1 Appendix Section A for more detail.) Below we describe the parameters used to model the physical process that yields these intensities:

**The rolonies, F**. The transcripts in the tissue are amplified in place into a 'rolony' structure which is easy for fluorophores to bind to [7]. Each voxel $m$ may contain a different amount of rolony material, and hence a varying level of fluorescence signal. We refer to the amount of material in voxel $m$ for rolonies associated with barcode $j$ as the **rolony density**. We denote this density by $\mathbf{F}_{m,j}$. The variable $\mathbf{F}$ indicates where rolonies are and how bright we should expect them to be. This density should always be non-negative. Note that $\mathbf{F}$ cannot be observed directly—instead, we observe fluorescence signal in different rounds and channels, and must use these signal observations to estimate the rolony densities.

**The codebook, B**. In each imaging round $r$, the rolonies associated with gene $j$ will bind to specific fluorescently labeled detection probes. We use the binary variable $\mathbf{B}$ to indicate which imaging rounds and fluorescent probes each gene is associated with. Specifically, we let $\mathbf{B}_{r,c,j} = 1$ whenever a rolony with barcode $j$ should bind to a fluorescent probe associated with specific color-channel $c$ in imaging round $r$, and otherwise we let $\mathbf{B}_{r,c,j} = 0$. Here we assume $\mathbf{B}$ is known. The vector of values of $\mathbf{B}$ for a particular gene $j$ is known as the 'barcode' for that gene, and the collection $\mathbf{B}$ of all the barcodes is known as the 'codebook.'

**The probe response functions, K, $\varphi$.** If a probe centered at a particular voxel is illuminated with a particular wavelength, the probe will emit a certain amount of signal which we can record at the corresponding voxel. We may also observe dimmer responses at neighboring voxels, due to the possible spreading of the single point object in the optical system. We use a non-negative matrix **K** to denote the *point-spread function*, i.e., the typical fluorescence signal-levels produced at each voxel in the neighborhood of a probe. We use the matrix $\varphi$ to represent the responsiveness of each type of fluorescent probe to each wavelength; each element of this matrix lies in the range of [0, 1]. Here we assume that the number of types of fluorescent probes is the same as the number of color-channels measured (though this could be relaxed). We further assume that the voxel-resolution of the rolony density is the same as the voxel-resolution of the original images.

**Phasing, $\rho$.** A washing process is applied after each round of imaging. However, in practice this washing step may not completely remove all of the reagents from every voxel. This can result in a 'ghost' of one round appearing in the next rounds. For each color-channel $c$, we let $\rho_c \in [0, 1]$ indicate the fraction of activity which appears as a 'ghost' signal in the next round.

**Background, $a$.** The images we obtain may also include background fluorescence from the tissue. We assume that the background is constant across rounds. We model this effect using a non-negative per-voxel value $a_m$ for each voxel $m$.

**Per-round per-wavelength gain, $\alpha$, and baseline, $b$.** The brightness observed from all rolonies at a particular color-channel in a particular round may have an associated gain factor. We model this gain factor with a non-negative per-round ($r$) per-channel ($c$) multiplier $\alpha_{r,c}$ and non-negative intercept $b_{r,c}$.

Putting all these pieces together, we obtain an *observation model*. This model states that the observed brightnesses $\mathbf{X}_{m,r,c}$ should be given by the formulae

$$\mathbf{X}_{m,r,c} \approx a_m + b_{r,c} + \alpha_{r,c} \sum_{j,m',c'}^{J,M,C} \mathbf{K}_{m,m'} \mathbf{F}_{m',j} \varphi_{c,c'} \mathbf{Z}_{r,c',j},$$

$$\mathbf{Z}_{r,c,j} = \rho_c \mathbf{Z}_{r-1,c,j} + \mathbf{B}_{r,c,j}.$$

Here the variable **Z** is used to incorporate the round-phasing effects; i.e., $\mathbf{Z}_{r,c,j}$ measures the concentration of probes of type $c$ which we would expect at round $r$, arising from a rolony with barcode $j$. We will also find it convenient to define

$$\mathbf{G}_{r,c,j} = \alpha_{r,c} \sum_{c'} \varphi_{c,c'} \mathbf{Z}_{r,c',j}.$$

This represents the total contribution of fluorescence signal expected to arise in round $r$ and channel $c$ from a rolony of type $j$. A summary of notation can be found in Table 1.

Overall, the model introduced above could certainly be expanded to model the physical imaging process more accurately, but we found that it was sufficient for our purposes: detecting and demixing rolonies.

## Inference

Our task is to uncover the positions and barcodes of rolonies in the tissue. According to the model in the previous section, this information can be obtained from the rolony density variable, **F**. However, **F** cannot be directly measured; thus our primary task is to estimate **F** from

**Table 1. Notation.**

|  | Description | Dimensions | Support |
|---|---|---|---|
| $M$ | number of voxels | scalar | $\mathbb{N}$ |
| $C$ | number of types of probes/wavelengths | scalar | $\mathbb{N}$ |
| $R$ | number of rounds | scalar | $\mathbb{N}$ |
| $J$ | number of barcodes | scalar | $\mathbb{N}$ |
| $\mathbf{X}$ | observed imaging intensities | $M \times R \times C$ | $\mathbb{R}^+$ |
| $\mathbf{F}$ | rolony density | $M \times J$ | $\mathbb{R}^+$ |
| $\mathbf{B}$ | (known) binary codebook matrix | $R \times C \times J$ | $\{0, 1\}$ |
| $\rho$ | per-channel phasing factor | $C$ | $[0, 1]$ |
| $\mathbf{Z}$ | phased barcodes | $R \times C \times J$ | $\mathbb{R}^+$ |
| $\alpha$ | per-round per-channel scale factor | $R \times C$ | $\mathbb{R}^+$ |
| $\mathbf{G}$ | scaled color-mixed phased barcodes | $R \times C \times J$ | $\mathbb{R}^+$ |
| $b$ | per-round per-channel offset | $R \times C$ | $\mathbb{R}^+$ |
| $a$ | per-voxel baseline intercept term | $M$ | $\mathbb{R}^+$ |
| $\mathbf{K}$ | spatial point-spread function | $M \times M$ | $\mathbb{R}^+$ |
| $\varphi$ | probe wavelength-response matrix | $C \times C$ | $[0, 1]$ |
| $\omega$ | tolerated reconstruction error | scalar | $\mathbb{R}^+$ |

the original image data. To do this we must in a sense invert the observation model specified above: the observation model tells us how rolony densities give rise to the fluorescence signal, but we would like to use observations of the fluorescence signal to estimate the rolony densities.

**Using the observation model to estimate the rolony densities F.**   We use a sparse non-negative regression framework to estimate the unknown parameters. In this estimation we are guided by three ideas:

- We believe our observation model is *approximately* correct. We formalize this by saying that we believe our squared 'reconstruction loss' can be made small. We define this loss by

$$L_{\text{reconstruction}} = \sum_{m,r,c} \left( \mathbf{X}_{m,r,c} - \left( a_m + b_{r,c} + \alpha_{r,c} \sum_{j,m',c'}^{J,M,C} \mathbf{K}_{m,m'} \mathbf{F}_{m',j} \varphi_{c,c'} \mathbf{Z}_{r,c',j} \right) \right)^2 .$$

- We believe that all of our parameters are non-negative. For example, we do not believe it is possible to have *negative* densities for rolonies at a particular voxel. Likewise, we expect the per-round per-channel scaling factors ($\alpha$) and probe-response terms ($\varphi$) to be non-negative.

- We believe that the rolony densities, **F**, are sparse: many voxels will not contain any rolony at all. Ideally we would formalize this idea by putting a penalty on the number of voxels with nonzero rolony amplification. However, this penalty is difficult to optimize in practice. Instead, following a long history of work in sparse estimation theory [16], we enforce this sparsity by placing a linear penalty on the total summed density. We define this penalty by

$$L_{\text{sparsity}} = \sum_{m,r,c,c',j} \alpha_{r,c} \mathbf{F}_{m,j} \varphi_{c,c'} \mathbf{Z}_{r,c',j}.$$

(Note that for a general sparse estimation problem, this penalty would be defined using a summed absolute value term; however, in our case all parameters are already constrained to be non-negative, so this is not necessary.)

Together, these three ideas suggest constrained optimization as a natural way to estimate our parameters. We will seek the non-negative parameters that give the smallest possible value of $L_{sparsity}$, subject to the constraint that $L_{reconstruction}$ falls below a noise threshold $\omega$. We provide an automatic way to select this noise threshold (see S1 Appendix, Section L), as well as an interactive process for the user to select this threshold so that the reconstruction loss appears satisfactory.

Assuming that **B**, **K** are known, this constrained optimization problem can be written as:

$$\min_{\mathbf{F},\rho,\alpha,b,a,\varphi \geq 0} L_{sparsity},$$

$$\text{subject to} \quad L_{reconstruction} \leq \omega. \tag{1}$$

To solve this optimization problem, we use a projected gradient descent approach. The quadratic structure of the problem makes it possible to pick all learning rates automatically, yielding fast convergence; for example, the resulting algorithm reaches convergence for a single $1000 \times 1000$ field of view (with a total of 28 images, with seven rounds and four color-channels) and 81 different barcodes (79 from the original experiment, and two additional unused barcodes as described below) in about two minutes on a `p2.xlarge` Amazon GPU instance. Details can be found in S1 Appendix, Section L.

Before concluding this section, we will address an issue of what is known as 'identifiability.' Let us say we have learned a model via our inference method, i.e. we have learned **F**, $\rho$, $\alpha$, $b$, $a$, $\varphi$. Now let us consider a new model, $\mathbf{F}'$, $\rho'$, $\alpha'$, $b'$, $a'$, $\varphi'$, such that

$$\mathbf{F}' = 4\mathbf{F} \qquad \rho' = \rho$$
$$\alpha' = \alpha/2 \qquad b' = b$$
$$a' = a \qquad \varphi' = \varphi/2.$$

Under this new model, the reconstruction loss is the same and the sparsity loss is the same. As far as our inference method is concerned, the two models are identical. It follows that our inference procedure simply cannot hope to learn overall scaling factors of this kind. Thus, any learned parameters should be understood as being known up to overall scale factors. To resolve this ambiguity we normalize $\alpha$ by dividing by its sum (recall that $\alpha$ is non-negative, so this sum will be positive) and multiply **F** by the same factor. Similarly, we divide each row of $\varphi$ by its diagonal value and multiply the corresponding column of $\alpha$ by the same value.

**Finding rolonies.** Let us now assume we have used the non-negative regression framework to estimate **F** (the collection of rolony density images, one for each barcode). These per-barcode density images indicate the positions of rolonies that belong to a particular barcode; see the left side of Fig 1 for a schematic. We can then apply a blob-finding algorithm to these per-barcode images to find the rolonies for each barcode; in practice we simply find local maxima in the per-barcode images.

Finding rolonies, or 'blobs,' in the per-barcode images is easier than finding blobs in the original images. See Fig 2A as an example. The per-barcode images include fewer blobs and the blobs are smaller, so there are fewer problems with overlapping blobs. More specifically:

**The rolony densities are demixed**. There are fewer blobs in each rolony density than in the original image stack. In the observed images, the intensity measured for each voxel for each wavelength at each round is a sum of contributions from all nearby rolonies which emit signal at that wavelength in that round. By contrast, the intensity measured at a particular voxel in the per-barcode images is only the sum of contributions from rolonies with that one specific barcode.

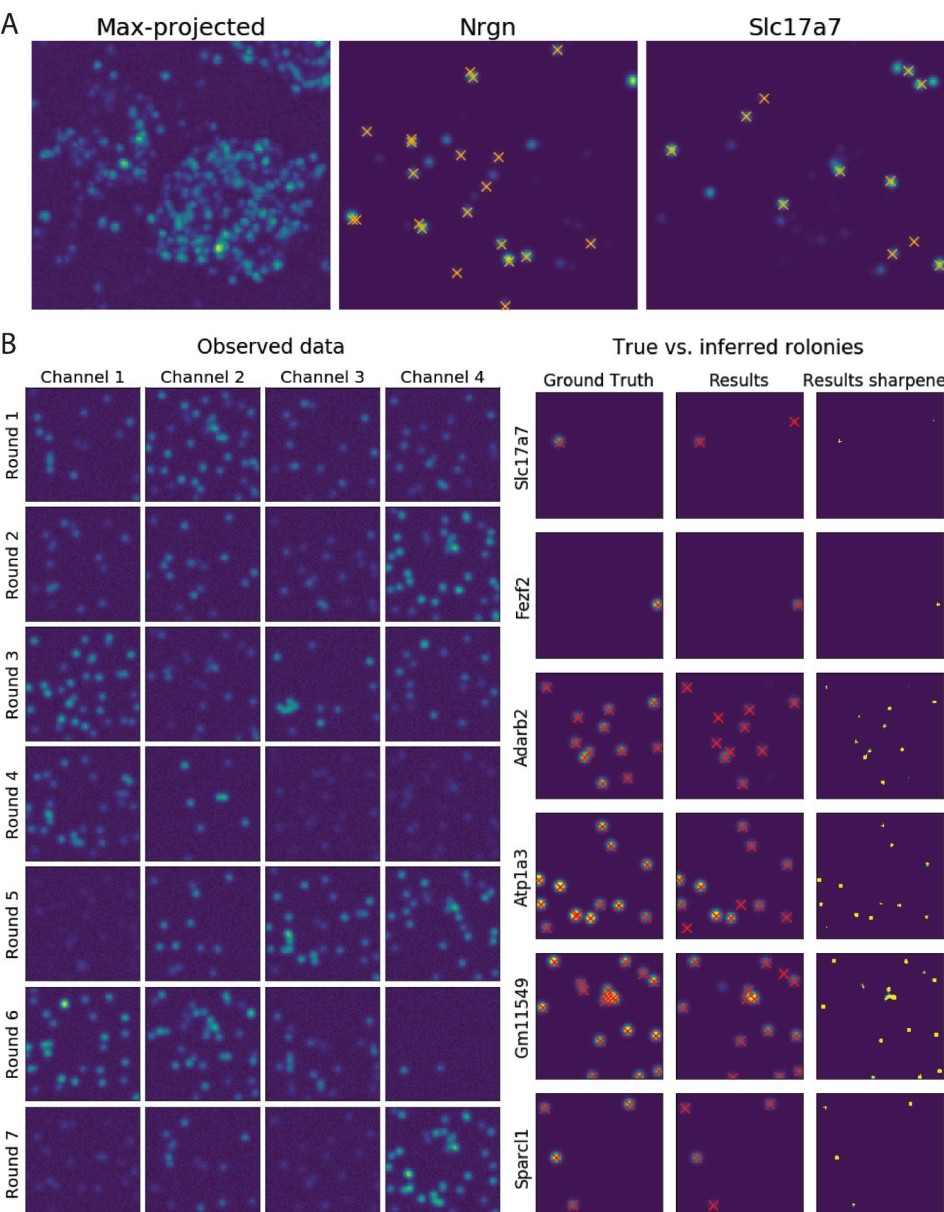

**Fig 2. BarDensr accurately estimates sparse, per-gene rolony densities from the proposed observation model. (A) Rolony densities make it easier to detect rolonies**. The left plot shows the max-projection of the original experimental image across all rounds and channels; detecting blob-like structures in this image can be challenging, especially when two rolonies are in close proximity. By contrast, the rolony densities for particular genes are sparser, so it is easier to identify the positions of individual rolonies in the tissue. The middle and right plots show examples of these rolony densities. The orange marks represent rolonies detected by a hand-curated approach. Note that the rolony densities appear to show several rolonies which were missed by the hand-curated approach (see S3 Fig for further details). **(B) BarDensr accurately recovers the ground truth in simulated data**. The left plot shows the simulated data in all rounds and channels. In the right plot, we applied BarDensr to this simulated data, and found that we were able to largely recover the true rolonies in this simulation (shown on the first column). The final column of plots shows the rolony densities learned by BarDensr, which shows that the algorithm accurately recovers most of the simulated ground truth rolonies, with a few mistakes. The middle column of plots shows a blurred version of the rolony densities and the spots discovered from these rolony densities.

**The rolony densities are deconvolved**. The blobs are smaller in the rolony density than in the original image stack. In the observed images, the intensity at a voxel is a contribution from all rolonies which are within the radius of the point-spread function **K**. Recall that this function smears signal from a single voxel across all nearby voxels. By contrast, the intensity of a per-barcode image at a particular voxel represents the amplification level of rolonies in that one voxel. In this sense, the inference process attempts to invert the point-spread function (i.e., perform deconvolution). On its own, this inversion process would not be numerically stable; however, the sparsity penalty and non-negativity constraint ensure it is numerically well-behaved [16].

The rolony density **F** thus represents a demixed and deconvolved version of the raw data. The original data is mixed, insofar as each raw intensity represents contributions from many barcodes. It is also convolved, insofar as each raw intensity represents contributions from many positions in space via the point-spread function. The non-negative sparse regression allows us to simultaneously demix and deconvolve, yielding per-barcode images which are cleaner and easier to understand.

Although it is easier to find blobs in the rolony densities, there is still one obstacle to be overcome: the threshold. Any blob-finding algorithm must specify an intensity above which a blob is considered real. How can this threshold be chosen? Here we make use of 'unused barcodes.' There could be as many as $C^R$ unique barcodes in a codebook for an experiment with $R$ rounds and $C$ channels of measurement (assuming only one channel emits signal in each round, which is the case in the experiments we studied). However, most of these barcodes are not used in the actual experiment. These unused barcodes give us a way to pick a sensible threshold. Along with the real codebook, we additionally include several unused barcodes; we enumerated all possible barcodes such that each round contained exactly one active channel, then selected uniformly at random from the set of barcodes such that each barcode differed from every other barcode in at least three rounds. We then run BarDensr on this augmented codebook. Blobs in the rolony densities associated with the unused barcodes must correspond to noise, since the true data-generating process did not include any signal from such barcodes. We therefore set the threshold to be the smallest value which guarantees that no blobs were detected in the unused barcodes. (In practice, using just two unused barcodes sufficed to estimate a stable and accurate threshold.)

**Accelerating computation.** The time required to apply BarDensr scales roughly linearly with the number of voxels in the data. There are several approaches the BarDensr package uses to relieve the computational burdens of working with large datasets:

1. **Exploiting barcode sparsity**. In any given patch of the data, many of the barcodes may not appear at all. If we can use a cheap method to detect genes which are completely missing from a given patch, we can then remove these genes from consideration in that patch, yielding faster operations. We call this 'sparsifying' the barcodes.

2. **Coarse-to-fine**. As we will see below, BarDensr is effective even when the data has low resolution. This suggests a simple way to accelerate computation: downsample the data, run BarDensr on the downsampled data (which will have fewer voxels), and then use the result to initialize the original fine-scale problem. If this initialization is good, fewer iterations of the optimization will be necessary to complete the algorithm.

3. **Parallelization**. BarDensr can use multiple CPU cores or GPUs (when available) to speed up parallel aspects of the optimization (e.g., processing data in spatial patches).

Details on these methods (which can be used in combination with each other) can be found in S1 Appendix, Section K.

## Code availability

The BarDensr Python package is available from https://github.com/jacksonloper/bardensr. The NeuroCAAS implementation of BarDensr can be found at http://www.neurocaas.com/analysis/8. This NeuroCAAS implementation requires no software or hardware installation by the user. As input, the user must upload a stack of images, a codebook, and a configuration file specifying parameters such as the radius of the smallest rolonies of interest (see S1 Appendix, Section C, as well as the NeuroCAAS link above for further details regarding the data format.) We assume that the images have been registered and background-subtracted before input into NeuroCAAS. There are three outputs from BarDensr NeuroCAAS implementation:

1. The first output indicates candidate rolony locations. This output is represented in a comma-separated-value file listing all entries in the rolony density **F** which have signal greater than zero.

2. The second output is designed to help the user assess the quality of the results; it is an HDF5 file which stores diagnostics about each detected rolony. See the next section for details.

3. The third output is a plot designed to help the user assess the overall quality of the results. For each detected rolony, we estimate the quality of evidence for that rolony by taking a key correlation coefficient (see the next section for details). The plot shows the histogram of these quality measures over all spots.

For further details on the AWS hardware used in the NeuroCAAS implementation, see S1 Appendix (Section D).

## Results

### The rolony densities estimated by BarDensr provide sparse, single images to detect spots for individual barcodes

As described in the Methods Section, the sparse non-negative regression approach aims to yield per-gene rolony density images which are easy to work with. The cartoon in Fig 1 may help illustrate this idea. Our belief is that the true per-gene rolony densities will be sparse images, so the learned rolony densities should also be sparse images.

To test this belief, we applied BarDensr to the experimental data described in the Methods Section. Fig 2A compares the raw data with the learned rolony densities for *Nrgn* and *Slc17a7* in a small region of the tissue. As hoped, the rolony densities are indeed quite sparse compared to the raw data. This ensures that blob-detection is relatively easy. This figure also shows that many of the bright spots in the rolony density images appear at rolony locations found by a hand-curated method (see S1 Appendix, Section E for details). For visualization purposes, Fig 2A shows the blurred version of the rolony densities (i.e. **KF**); these make it easier to see the bright spots.

To get a sense for what all the different genes look like, we also examined the rolony densities for all the barcodes (81 in total in this dataset, including two unused barcodes); see S1 and S2 Figs. These sparse images enable us to identify the rolony location easily for each barcode.

## BarDensr provides improved demixing and detection accuracy compared to existing approaches

To benchmark BarDensr against other methods, we generated simulated data with rolony density, gene expression levels, and noise levels matched to the experimental data, as shown in Fig 2B. Then, we examined how well we could recover the 'true' rolonies from the simulation data. Qualitative results for several different genes are shown in S4 Fig. Quantitatively, we present a Receiver Operating Characteristic curve (ROC curve) in Fig 3A, which summarizes the percentage of true detected rolonies (also known as '1-FNR', the complement of the False Negative Rate (FNR)). Depending on the False Positive Rate (FPR) we are willing to tolerate, different detection rates can be achieved; the ROC curve summarizes this relationship.

We compare BarDensr to several other approaches. *Starfish* is one package developed for analyzing spatial transcriptomics data. This method has many hyperparameters. To give this method its best chance, we first tried to find the best parameters manually, and additionally used the *BayesianOptimization* package [17] to find the hyperparameters which allowed it to perform as well as possible on the simulated data. Fig 3A shows that this performance falls short of the detection rates achieved by BarDensr. We also investigated SRM (see S1 Appendix, Section E) and a correlation-based method ('corr', see S1 Appendix, Section F) for

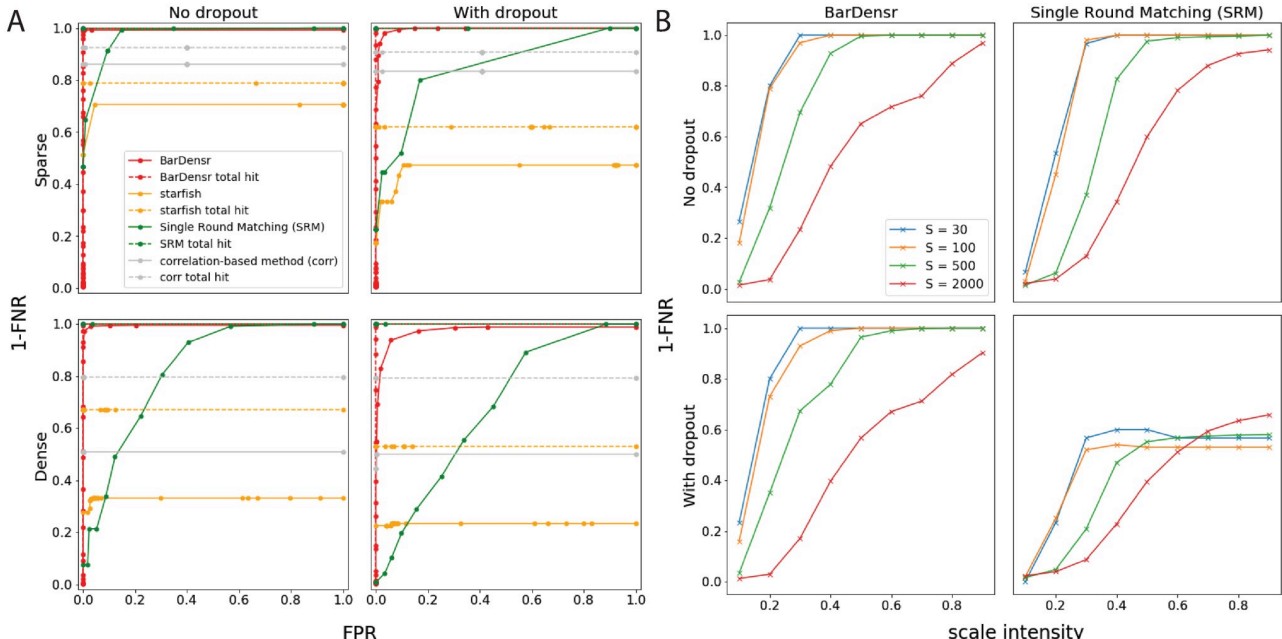

**Fig 3. BarDensr outperforms the existing methods by discovering more rolonies. (A) Performance of BarDensr on simulated data**. What percentage of rolonies are correctly detected? We use the Receiver Operating Characteristic curve (ROC curve) to look at this percentage (the complement of False Negative Rates, or 1-FNR) as a function of the tolerated False Positive Rate (FPR), for BarDensr (red), *starfish* (orange), Single Round Matching (SRM, green), as well as the correlation-based method ('corr', gray); cf. S1 Appendix, Sections E and F for details on these other methods. S4 Fig illustrates these simulation data. In drawing these curves, we consider two qualitatively different kinds of errors: errors because a rolony isn't detected at all (dotted lines), and errors because a rolony is detected but it is assigned the wrong barcode (solid lines). The left plots show these curves for simulated data. The right plots show these curves for simulated data with 'dropout'—a form of noise present in some spatial transcriptomic methods (cf. S1 Appendix, Section G for details). For all four kinds of simulations, we found BarDensr is able to find significantly more spots. **(B) Performance of BarDensr on the hybrid simulation**. Simulated data is always imperfect; to try to measure performance on a more realistic dataset, we used a hybrid method a la [18]. We injected fake rolonies into real data, and quantified how well different methods could recover these fake spots. The plots above show 1-FNR (y-axis) as the function of scale intensity of the fake rolonies (x-axis) and number of fake rolonies injected (*S*, colored lines), without (top) and with (bottom) dropout, using BarDensr (left) and SRM (right). See S1 Appendix, Section G for details.

comparison. These two methods represent 'blobs-first' and 'barcodes-first' approaches. Bar-Densr has better recovery prediction than either of these methods.

Our simulated data here do not capture the full biological content of the real observed data. For example, in real data, the tissue often has some regions with dense rolony concentrations (e.g. nuclei) and other regions which are more sparse. In order to quantify performance in more realistic biological contexts, we performed a 'hybrid' simulation, a la [18]. We started with the original experimental data and injected varying numbers of spots at random locations in the image with varying peak intensities (cf. S1 Appendix, Section G). To test if the model is able to recover these injected spots with the original image background, we computed 1-FNR (FPR could not be computed here since we do not know the ground truth in the original experimental data). We ran two variants on this simulation: one ordinary simulation and one simulation with 'dropout,' in which some rolonies emit a strong bright signal in most of the rounds but simply vanishes in one or more rounds (see S1 Appendix, Section G). The results of the dropout and non-dropout experiments are shown in Fig 3B. As expected, the performance decreases when the intensity of the injected spots is smaller. However, as long as the intensity of injected spots was at least half the maximum intensity of the original image, and the number of injected spots is in a reasonable range, BarDensr was able to find all the spots, even in the simulation with dropout; by contrast, the SRM approach was unable to find all the injected spots in the hybrid experiment, especially in the dropout variant.

## Errors are mostly mis-identification on the barcodes, not missed detections

We used simulated data to investigate the failures represented by the FPR and FNR described above: are they caused by failure in assigning the rolonies to the correct barcodes ('barcode mis-identification'), or failure in detecting rolonies? To find out, we computed how the failure rates would change if mis-identified barcodes were not considered 'errors.' We denote this the 'total hit rate' analysis (cf. Fig 3A, dotted lines); both BarDensr and SRM have very high total hit rates for the simulated data examined here, indicating that both of these methods detect spots well, but sometimes mis-classify the spot identity. See S1 Appendix (Section H) for further details.

## BarDensr remains effective on data with low spatial resolution

High-resolution imaging can be expensive and time-consuming. BarDensr can also work on low-resolution images. To show this, we spatially downsampled the experimental images for each frame (each round and each channel). We then fit BarDensr to these lower-resolution images. An example is shown in Fig 4 (additional examples with 5× and 10× lower resolutions can be seen in S5 Fig). These figures suggest that BarDensr correctly detects the overall expression levels of each gene in low-resolution images—even when the downsampling is so extreme that picking out individual rolonies is not feasible.

To more rigorously test if BarDensr can recover the correct gene expression level when applied on the low-resolution data, we quantified the cell-level gene activity on a larger region where 43 cells are detected using a seeded watershed algorithm (see S1 Appendix, Section I for detail). Fig 4C suggests that with 5× downsampled data, the cell-level gene expression, as well as the cell clusters, are preserved with high consistency compared to the results of applying the method to the original fine scale.

Finally, we wanted to test if BarDensr can correctly identify individual rolonies on a low-resolution version of the experimental data from a recent study [14]. There was no ground truth available for this real data. We therefore manually inspected every location in a small

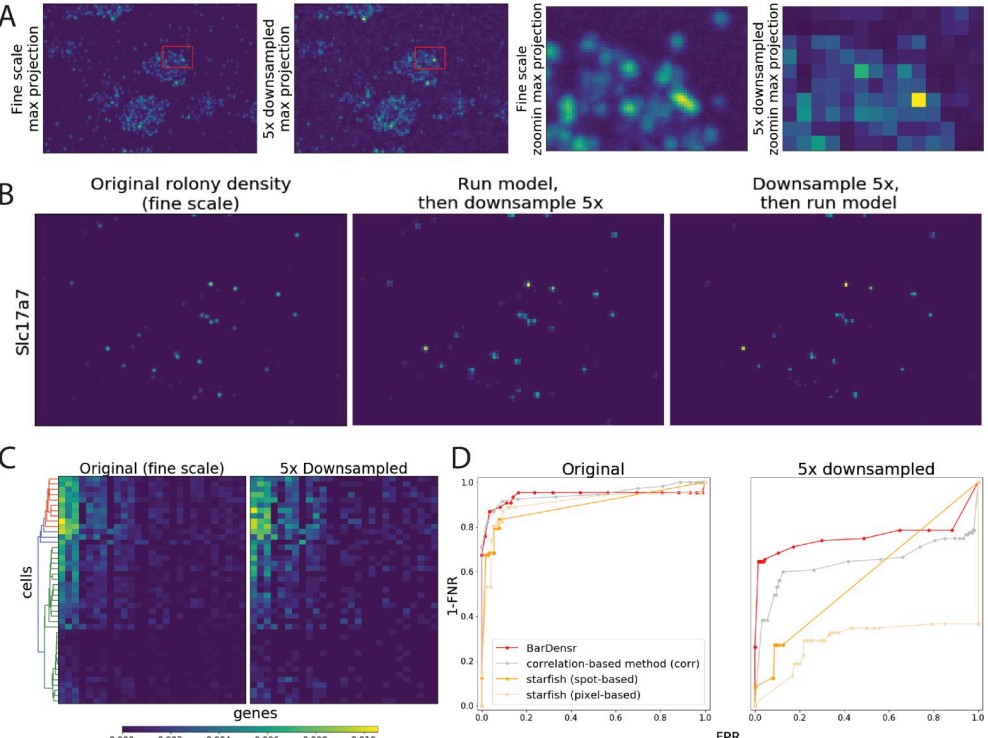

**Fig 4. BarDensr is the best choice for dense and/or low-resolution experimental data. (A)** The 5× downsampled image is compared to the original 'fine scale' image. All these plots show the max-projection across all rounds and channels, with the right two showing the zoomed region indicated by the red rectangles in the left two. Note that it is difficult to visually isolate single spots from the downsampled image. To test the performance of BarDensr on this low-resolution data, we first run the model on the original data, obtain rolony densities, and then finally downsample the rolony densities. Next, we run BarDensr on downsampled data, and examine the estimated rolony densities. **(B)** The rolony densities for a selected gene (*Slc17a7*) estimated using the original fine scale (left), as well as these two approaches. For a more complete example, see S5 Fig. **(C)** The cell-level gene expression quantification, for those genes that have more than four spots in the fine scale in a 1000 × 1000 region. The color of the heatmap indicates the proportion of gene counts (i.e., the total counts of each gene divided by the total counts of all genes detected in the region). The x-axis represents the 24 genes that were chosen, ordered based on the counts in the fine scale. The y-axis represents the cells, ordered based on the hierarchical clustering result from the fine scale, as shown in the dendrogram on the left. A total of 43 cells are segmented from the original image using a seeded watershed algorithm (cf. S1 Appendix, Section I). The two different results yield nearly identical clusterings, indicating that BarDensr recovers gene activity with accuracy sufficient to cluster cells even given low-resolution images. **(D)** In order to evaluate the performance of BarDensr and compare it with the other state of the art methods, we took a 200 × 200 region of real experimental data from a recent study [14] and created a benchmark dataset (cf. S1 Appendix, Section J). Above we plot the ROC performance on this benchmark for four different methods (BarDensr, correlation-based method, as well as *starfish* with 'spot-based' and 'pixel-based' approaches), using the original data and denser / relatively low-resolution data. While all the methods performed quite well on the original images (left), BarDensr has a better performance than the others when the lower-resolution images were used (right).

region (200 × 200 voxels) to identify all of the rolonies and created a benchmark dataset. We compared the performance of BarDensr, the correlation-based method, and *starfish* with 'spot-based' (the method that is recommended for this data based on *starfish*) and 'pixel-based' methods. The left plot in Fig 4D shows that all these four methods perform quite well at the original resolution, with the correlation-based method and *starfish* performing faster compared to BarDensr (0.59 seconds for the correlation-based method; 1.22 and 8.16 seconds for *starfish* pixel-based and spot-based method, respectively; BarDensr used 21.77 seconds). When the original images were 5 times blurred and downsampled to emulate denser data, as shown on the right plot of Fig 4D, BarDensr recovered more correct rolonies compared to the other

methods. For experiments with dense and/or low-resolution images, these results (and the simulation results above) suggest that BarDensr is the best available analysis tool. More details on how we generated the benchmark can be found in S1 Appendix (Section J).

## BarDensr computations can be scaled up to large datasets via sparsifying and coarsening accelerations

In the Methods Section, we described how the barcode sparsity could help us potentially apply the method to a large dataset with more barcodes. To test if we can use a much larger dataset, we considered a simulated example with more unique barcodes (53,000 unique barcodes and 17 sequencing rounds). With so many barcodes, naively running BarDensr is prohibitively expensive (in both compute time and memory) on large datasets. However, we also expect such datasets are extremely sparse in terms of barcodes—any given small region of the image is quite unlikely to include rolonies from all 53,000 barcodes. This is particularly true for some of the latest applications of multiplexed imaging technologies, in which each unique barcode corresponds to dendrites from a unique cell, instead of each barcode corresponding to a gene [13]. In these experiments, a small region of tissue may contain many different transcripts, but it will only contain portions of a small number of different cells. Thus we should be able to take advantage of this sparsity to speed up BarDensr. We simulated a $50 \times 80$ small region where 40 rolonies were present in total. We then obtained a coarse, downsampled image, and then ran BarDensr and learned the parameters for this low-resolution data. If the learned parameters from the coarse scale indicated a particular barcode did not appear, then we assumed that this barcode should be absent even if we used the data at the original resolution. The result in S6 Fig shows nearly perfect prediction performance. This problem was quite small, so we could also run the method without using any sparsity-based acceleration techniques; we found that the unaccelerated version did not outperform the accelerated version, suggesting that BarDensr can be used for datasets of this kind with a larger number of molecular or cellular barcodes (cf. [13, 19–21]).

When the number of barcodes is small, BarDensr can run without these acceleration techniques—but these accelerations may still be worth applying, to help cut down on computation times and reduce memory usage. As shown in Fig 5A, we found that these techniques reduced runtime by a factor of four in simulated data involving 79 barcodes (see S1 Appendix, Section K for details). Fig 5B further shows that these acceleration techniques do not significantly alter the results, even on real data.

Making use of these acceleration approaches, we applied BarDensr to a large section of mouse motor cortex from a recent study [14]. As shown in Fig 6, BarDensr was able to reveal the striation of the cell type marker genes and identify cell types consistent with known markers. The entire region shown is composed of 35 fields of view (each field of view is of size $2048 \times 2048 \times 1$ voxels), and 65 genes are targeted. The entire process with the usage of acceleration approaches takes approximately 3 hours to compute on a GPU machine (a `p3.2xlarge` machine on the AWS service).

## BarDensr recovers interpretable parameters

BarDensr uses a data-driven approach to estimate all the relevant features of the physical model: the per-channel phasing factor, the per-round per-channel scale factor, the per-round per-channel offset, the per-voxel background, the per-wavelength response matrix, and the rolony densities (the latter of which have already been described in detail above). In the data analyzed here, we found that the per-channel phasing factor was relatively small, suggesting very little 'ghosting' in this data. The wavelength-response matrix was almost diagonal,

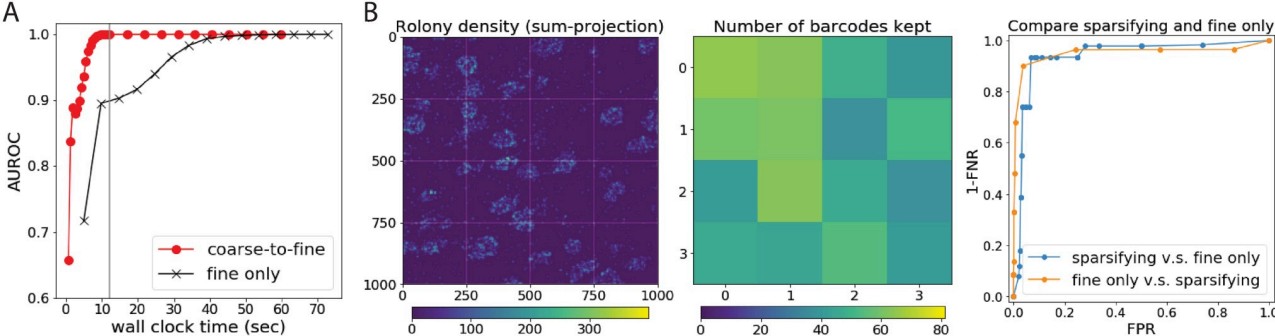

**Fig 5. BarDensr can be scaled up with sparsifying and coarse-to-fine approaches. (A) Coarse-to-fine acceleration**. The Area Under the ROC (AUROC) summarizes the performance of a method by calculating the integral of the ROC curve (higher is better). The black curve plots the AUROC performance on the simulated data, against the number of seconds the default iterative algorithm has been allowed to use, up to a maximum of 15 iterations. We can also use a coarse-to-fine strategy, where we first run the algorithm on downsampled data for 20 iterations, and use the results to perform 10 additional iterations on the full high-resolution data; the red curve plots the performance for this strategy. **(B) BarDensr can take advantage of gene-sparsity**. Here we used two different approaches to analyze a 1000 × 1000 region of the experimental data. The first approach uses BarDensr naively, applying it directly to the image. The second approach, illustrated on the first two plots, accelerates the method using a 'coarse-to-fine' method by taking advantage of 'gene-sparsity.' Specifically, we split this region into 4 × 4 patches (the borders of these patches are indicated as the white lines on the left plot). After the relatively fast 'coarse' step, the barcodes that have very low maximum rolony densities were removed before the following 'fine' step. This keeps only a relatively small number of barcodes to consider for each patch (ranging from 38 to 65 out of 81 barcodes, as shown in the middle plot), therefore reducing the computation time and the memory usage for the 'fine' step later (cf. S1 Appendix, Section K for more detail). We here show that both methods yield nearly the same result, as shown in the ROC curves on the right plot. In particular, we treated one method as the 'truth' and constructed an ROC curve indicating the accuracy of the other method. We can then do the reverse, treating the other method as 'truth.' The results suggest strong agreement.

although we found some slight color-mixing from channel 2 to channel 1, consistent with visual inspection (see the fifth round in S7(B) Fig as an example). This indicates that our model is able to correctly recover the color-mixing effects. We also investigated whether all of the features of our model were necessary for the purposes of finding rolonies. For each feature of the model, we tried removing that aspect of the model and seeing whether the method still performed well. For the data analyzed here, we found that the $\varphi$ and $\rho$ parameters were not essential (though they did seem to improve the performance, at least qualitatively). By contrast, all of the other parameters were essential; removing any of them yielded nonsensical results.

## The BarDensr model correctly predicts the observed signal intensities

This algorithm is based upon a physical model of how this data is generated. Rolonies appear at different positions in the tissue, they emit fluorescence signal in different conditions, the fluorescence signal is smeared by a point-spread function, and finally we observe this signal, together with certain background signal and noise. As long as this model captures all the important features of the physical process, observed intensities should match the predicted intensities at each voxel in each round and in each channel. To think about this more clearly, let's define these predicted intensities as the 'reconstruction':

$$\text{reconstruction}_{m,r,c} \triangleq a_m + b_{r,c} + \alpha_{r,c} \sum_{j,m',c'}^{J,M,C} \mathbf{K}_{m,m'} \mathbf{F}_{m',j} \varphi_{c,c'} \mathbf{Z}_{r,c',j}.$$

To test our model, we can visually compare the reconstruction to the observed data. If the residual between the two includes significant highly-structured noise, then it is likely that we are missing important aspects of the data. S7–S9 Figs show the results of these comparisons, and the overall reconstruction error is approximately 7.64% (computed as the percentage unexplained) in the 180 × 200 region shown in S7(A)–S9(A) Figs. (Also see the Supplementary

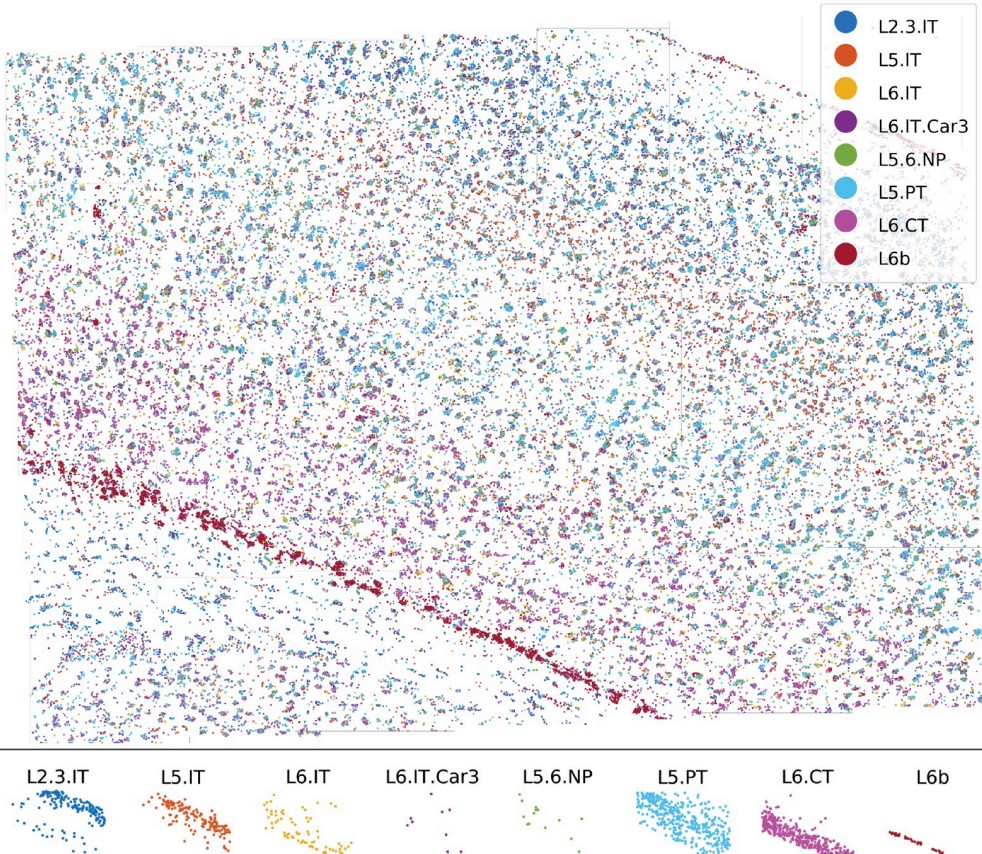

**Fig 6. BarDensr reveals the laminar distribution of the cell type marker genes, as well as the identified cell types in the motor cortex.** The data was obtained from a recent study [14]. **Top**: each dot indicates a detected rolony; for each cell-type, we use the same color for all rolonies associated with marker-genes for that cell-type. **Bottom**: each dot indicates a cell. Dots are colored based on the cell-types; cell-types are estimated using the detected rolonies from the top plot.

video, described in S1 Appendix, Section B.) They appear fairly promising, but certain structured features do appear in the residual. Most strikingly, we have found that a minority of rolonies 'dropout' for one or more rounds: a rolony may give a strong bright signal in most of the rounds but simply vanish in one round. Our current physical model does not accommodate this, and this limitation appears in the residual as bright and dark spots. However, as mentioned above and shown in the hybrid simulation data in Fig 3B, our method is robust to these 'dropout' effects; it is still able to capture the correct rolony positions when dropout occurs on a small number of rounds.

## Diagnostics based on 'cleaned' images are useful to check the accuracy of BarDensr

The reconstruction is made up of many parts: it has the background component $a$, the per-round per-channel offset and scale terms $(\alpha, b)$, and rolony contributions arising from $\mathbf{F}, \varphi, \mathbf{Z}$. As shown above, it is straightforward to compare the total reconstruction to the observed data. However, this does not isolate the contributions of individual estimated rolonies.

Therefore we adapted a partial subtraction approach from [22]. We pick one barcode, $j^*$, and focus only on the contributions to the reconstruction from this one barcode. In particular,

we assume that all other aspects of the model are exactly correct. We assume that $a$, $\alpha$, $b$, $\varphi$ and $\mathbf{Z}$ are all exactly right. We further assume that $\mathbf{F}_j$ is exactly correct for every $j \neq j^*$. Assuming all these aspects of the model were perfect, we can look at what the data *would have looked like* if it had only included one type of barcode, namely $j^*$. We call this counterfactual simulation the 'cleaned image':

$$\mathbf{X}_{m,r,c}^{(j^*)} = \mathbf{X}_{m,r,c} - a_m - b_{r,c} - \sum_{j \neq j^*, m'} \mathbf{K}_{m,m'} \mathbf{F}_{m',j} \mathbf{G}_{r,c,j}. \tag{2}$$

This is the data with all aspects of the model subtracted away—*except* for the contributions from barcode $j^*$ (see Fig 7A as an example). The cleaned image for the barcode $j^*$ has much in common with the rolony density for $j^*$. However, $\mathbf{X}^{(j^*)}$ differs from $\mathbf{F}_{j^*}$ in one crucial way. For each voxel $m$, $\mathbf{F}_{j^*}$ gives exactly one value. However, for each voxel $m$, $\mathbf{X}^{(j^*)}$ gives $R \times C$ values— one for each round and channel of the experiment. According to our model, however, it should be possible to express all these values in terms of multiplication in the following form:

$$\mathbf{X}_{m,r,c}^{(j^*)} \approx \mathbf{F}_{j^*,m} \times \mathbf{G}_{j^*,r,c}.$$

In this formula (known as an 'outer product') we see that $\mathbf{X}_{m,r,c}^{(j^*)}$ (which varies across voxels, rounds, and channels) is the product of two objects: the rolony density (which varies across voxels) and the transformed barcode $\mathbf{G}$ (which varies across rounds and channels) for $j^*$. This is actually a very strong assumption; most tensors would not exhibit this kind of structure. We can empirically check for this 'rank-one' structure by computing the singular value decomposition (SVD) of $\mathbf{X}_{m,r,c}^{(j^*)}$. If the SVD yields only one strong singular value, then $\mathbf{X}_{m,r,c}^{(j^*)}$ can be well-approximated by this rank-one outer product, and furthermore the SVD yields the correct values for $\mathbf{F}_{j^*,m}$ and $\mathbf{G}_{j^*,r,c}$. We can compare the values for these quantities (as returned by the SVD analysis) to the estimated values (as returned by BarDensr). We show some examples in Fig 7B comparing the estimated value of $\mathbf{G}_{j^*}$ with SVD results (a similar but more complete set of the spots can be seen in S10 Fig). Note that the match isn't quite perfect (the temporal singular vector of the corresponding cleaned images varies a bit from our estimate). In future work we hope to investigate whether these differences could be accounted for by a more accurate physical model. For now, we content ourselves that the method is accurate enough to provide a useful diagnostic for the detected rolonies.

We can also use these cleaned images to help us compare BarDensr with other methods by eye. S11 Fig investigates cleaned images for gene *Arpp19*, comparing the results of our method to the hand-curated results. In cases where the results of the two approaches disagree, these cleaned images suggest that our results are often reasonable.

## Discussion

By directly modeling the physical process that gives rise to spatial transcriptomics imaging data, we found that BarDensr can correctly detect transcriptomic activity—even when rolonies are densely packed in tissue or optical resolution is limited.

BarDensr is computationally scalable, but so far we have only investigated real-world transcriptomic experiments with less than a thousand barcodes. To scale to larger barcode libraries we need to address the possibility that the barcode library may be unknown or corrupted. In experiments with tens of thousands of barcodes, some barcodes present in the data may be unknown to the experimentalist. If these barcodes are ignored, the performance of our method may be negatively impacted. In the future we hope to adapt our method to learn these barcodes directly, using the model outlined in this paper. Together with the computational acceleration

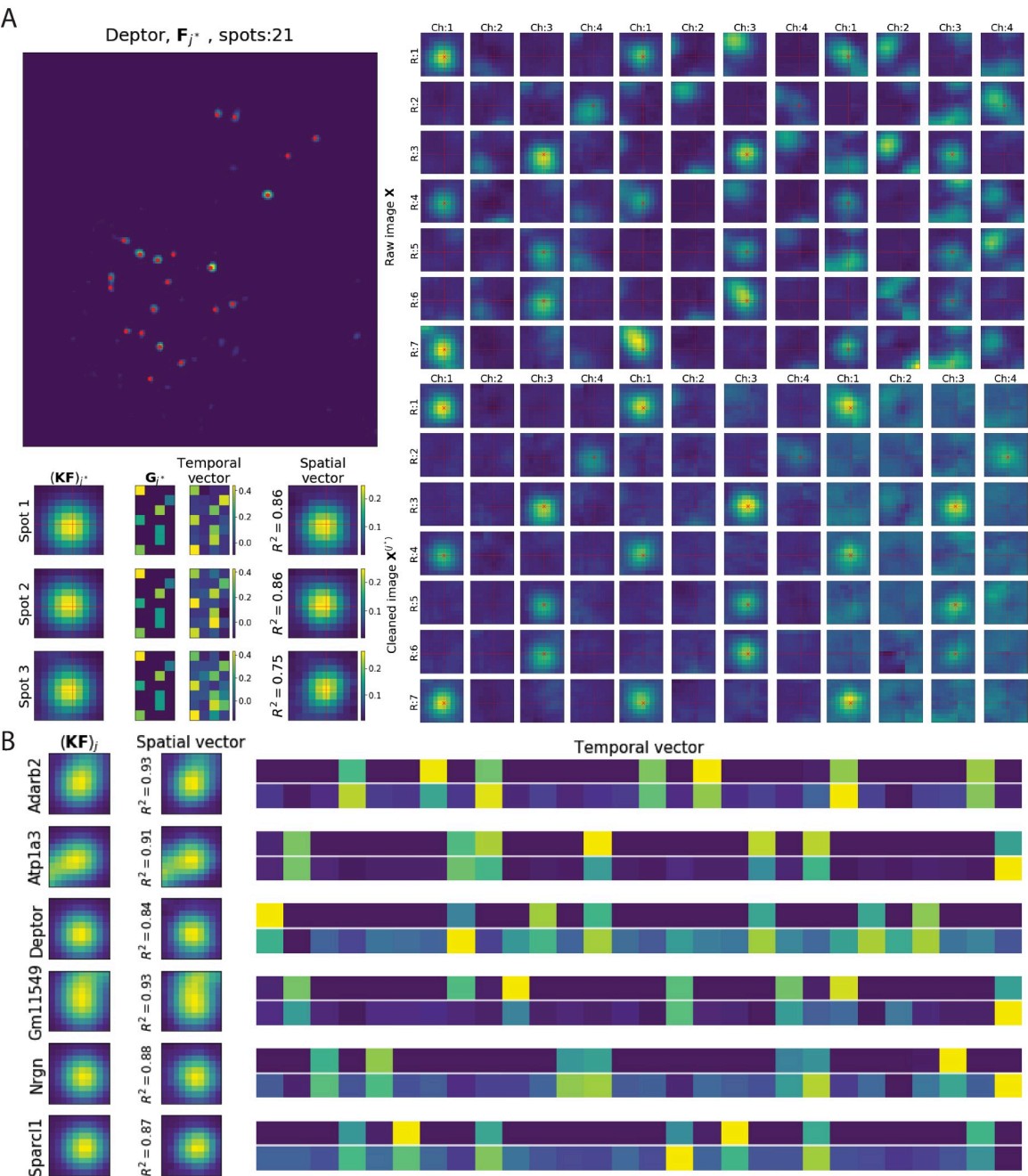

**Fig 7. Using cleaned images and SVD to examine model fit quality and variability. (A) SVD analysis example using one gene**. Spots are identified in $\mathbf{F}_{j^*}$ for each barcode $j^*$ using local-max-peak-finding. For the gene barcode $j^*$ (*Deptor*) shown here (top left), three spots with the highest accuracy are being analyzed. The right panel shows the zoomed-in $R \times C$ plots of the raw image $\mathbf{X}$ (top) and 'cleaned' image $\mathbf{X}^{(j^*)}$ (bottom) at these three spot locations for barcode $j^*$. Note that 'cleaned' images are significantly sparser than the raw images, as desired. We then applied SVD to the cleaned image $\mathbf{X}^{(j^*)}$ at these three spot locations. The first two columns on the bottom left show the zoomed-in image of the original spot $(\mathbf{KF})_{j^*}$ and the learned weighted barcode matrix $(\mathbf{G}_{j^*})$ corresponding to this gene barcode $j^*$. The top singular vectors are plotted in the last two columns (showing a good match with $\mathbf{G}_{j^*}$ and the cropped $(\mathbf{KF})_{j^*}$). $R^2$ is the squared correlation coefficient between $\mathbf{X}^{(j^*)}$ and the outer product of these two singular vectors; the high $R^2$ values seen here indicate that the model accurately summarizes $\mathbf{X}^{(j^*)}$. **(B) Results of SVD analysis of cleaned images for the top high-$R^2$ spots**. This plot summarizes the results of the analysis illustrated in (A). The first column shows $(\mathbf{KF})_{j^*}$ around the brightest spots; the second column shows the top spatial singular vectors for the same region, and the last column shows the top temporal singular vectors for these spots (the top row shows the scaled $\mathbf{G}_{j^*}$ learned from the model, and the bottom row shows the corresponding top temporal singular vectors for these spots). Only six barcodes that are most abundant in the selected region are shown here; S10 Fig provides a more complete illustration.

approaches used in this paper, this would extend BarDensr to larger-scale data with potentially corrupted barcode libraries.

There are several general challenges that are not addressed in our method. For instance, BarDensr assumes its inputs have been preprocessed appropriately, including background removal and registration across imaging cycles and channels. In future work it would be useful to incorporate these steps in a full self-contained processing pipeline.

Finally, we would also like to explore using amortized inference to accelerate BarDensr [23, 24]. The BarDensr algorithm must repeatedly solve the same kind of sparse non-negative regression problem throughout the tissue. It may be possible to train a convolutional neural network to directly solve these problems in a single step. If this proved feasible, it would also open the door to using more sophisticated nonlinear models to represent the physical processes that transform the unobserved rolony densities of interest into the noisy image stacks we observe. This could further enhance performance in the presence of e.g. dropout or other nonlinear artifacts.

## Supporting information

**S1 Appendix. Model development and analysis details.**
(PDF)

**S1 Fig. Rolony density (F)$_j$ of all 81 barcodes.** These images are the supplement to Fig 2A in the main text. The rolony densities represent a *demixed* view of the data. Each plot corresponds to a single barcode, and indicates the rolony density at different spatial locations. Above we show these rolony densities for one region in the experimental data. The title for the plots above indicates the gene associated with the barcode as well as the maximum intensity of the plot. The orange dots represent rolonies detected by a hand-curated approach.
(TIF)

**S2 Fig. Rolony density (KF)$_j$ of all 81 barcodes, after applying the point-spread function.** As of S1 Fig, these images are the supplement to Fig 2A in the main text, except we display (**KF**)$_j$ instead of (**F**)$_j$ for each barcode $j$. Recall that the point-spread function **K** has the effect of smearing signal over a spatially localized area. It represents physical processes which blur the signal of interest. Under the BarDensr model, the signal intensities observed at each voxel $m$ from a given barcode will arise directly from linear combinations of (**KF**)$_{m,j}$ over different barcodes $j$.
(TIF)

**S3 Fig. Supplement to Fig 2A. (A)** *Nrgn*. The top plot shows the same rolony density of *Nrgn* as in Fig 2A. The orange crosses indicate the spots detected in the hand-curated results. The three spots highlighted with red are further zoomed in the bottom. These spots were detected to have large signal intensities by BarDensr, but were not detected in the hand-curated results. The correct barcode frames for *Nrgn* are indicated with red crosses in the bottom plots, suggesting that each of these spots appear to be well-modeled as *Nrgn* spots. **(B)** *Slc17a7*. The top plot shows the same rolony density of *Slc17a7* as in Fig 2A. The orange crosses indicate spots detected by hand-curated method. The four spots highlighted with red or cyan are further zoomed in the bottom. The first three spots (Spot 1—3, shown in red) were found by BarDensr but were not detected by hand-curated results. The fourth spot (Spot 4, shown in cyan) is the spot that is detected by hand-curated results, but no signal detected in BarDensr. The correct barcode frames for *Slc17a7* are indicated with red crosses in the bottom plots.
(TIF)

**S4 Fig. (A) Benchmarking results on the regular simulation**. Comparing *starfish*, SRM, 'corr', and BarDensr results, to the ground truth. Showing the top three barcodes with highest density (the gene density was generated randomly, see S1 Appendix, Section G). The left panel is with no dropout, and it corresponds to the top left plot in Fig 3A in the main text. Without dropout, BarDensr accurately detects the barcodes in the original data. The right panel is similar to the left, but with dropout for 50% of the simulated spots. This corresponds to the top right plot in Fig 3A. **(B) Benchmarking results using five times denser simulation**. This is similar except that the spots density is five times denser than (A). The left and right panels are without and with dropout, as explained earlier, and they correspond to the bottom left and bottom right in Fig 3A, respectively. With dropout for 50% of the densely simulated spots, some missing spots (FN) can be observed from these methods (e.g., see the first row *Atp1a2*). False discovery (FP) can also be seen in this plot for SRM (e.g., see the third row *Rbfox3*).
(TIF)

**S5 Fig. Further demonstration of BarDensr applied to low-resolution data (supplementing Fig 4).** To test BarDensr's performance on low-resolution data, we first run BarDensr on the original data, obtain rolony densities, and then finally downsample the rolony densities ('run-then-downsample'). Next, we run BarDensr on downsampled data and look at the learned rolony densities ('downsample-then-run'). For highly-expressed genes, these two results are nearly indistinguishable.
(TIF)

**S6 Fig. BarDensr can be scaled up to a larger number of barcodes by using sparsified image.** To test if we can scale up BarDensr, we computed an ROC curve for the method using a simulated dataset with 53,000 barcodes and 17 sequencing rounds. After running the model on a 5× downsampled $50 \times 80$ voxels simulated image, barcodes that are set to zero at the coarse scale were removed and the model was run at the original scale, with the parameters learned from the downsampled image as the initial conditions. See also S1 Appendix (Section K).
(TIF)

**S7 Fig. Original data (X) after normalization for each round and channel.** In order to create clearer visualizations, we noise-normalized the data as described in S1 Appendix (Section A), so that images from all rounds and channels are on the same scale. (A) shows the zoomed-out images in the selected region. (B) shows the zoomed-in images for one of the target spots (a $20 \times 20$ region). Also see the video visualization at https://tinyurl.com/y7zzyrd4.
(TIF)

**S8 Fig. Data reconstructed from BarDensr.** Under the BarDensr model, the fluorescence signal observed at each voxel in S7 Fig. should be approximately given by the equations from the Methods Section. We here plot the results of those equations, visualized using the same color-map-intensity scale as used in S7 Fig. At least by eye, we see excellent agreement between the data and the model's predictions. (A) and (B) are zoomed-out and zoomed-in images as described in S7 Fig. Also see the video visualization at https://tinyurl.com/y7zzyrd4.
(TIF)

**S9 Fig. Residuals.** As mentioned in S8 Fig, the BarDensr model makes predictions about what the observed data should look like. There is broad agreement, but there is some disagreement. Here we highlight the the residual between the predictions and the data. Note the difference in scale compared to the previous two figures. (A) and (B) are zoomed-out and zoomed-in images as described earlier. Also see the video visualization at https://tinyurl.com/y7zzyrd4.
(TIF)

**S10 Fig. Further SVD analysis of cleaned images for the top high-$R^2$ spots.** The figure supplements Fig 7 in the main text, and is structured in the same way, except that this plot shows more examples (with more barcodes and spots). Each row shows two spots for a given barcode. The first two columns show $(\mathbf{KF})_{j*}$ cropped around the two spots; the third and forth columns show the top spatial singular vectors for the same crops. The final wide column shows the top temporal singular vectors for these spots, with the first row (above the thin white line) showing the scaled $\mathbf{G}_{j*}$ learned from the model, and the following two rows showing the corresponding top temporal singular vectors for these spots. The two spots are ordered by $R^2$, which is computed as in Fig 7.
(TIF)

**S11 Fig. Rolony density comparison against hand-curated results.** On the top two plots, we show the rolony density $\mathbf{F}_j$ (left) and the blurred rolony density $(\mathbf{KF})_j$ (right) for gene *Arpp19*, derived from the experimental data. These rolony densities indicate the presence of *Arpp19*-rolonies. However, they might be incorrect, indicating that these detected rolonies might not be present in the real data. In this figure we investigate this question qualitatively. First, we compare with the rolony positions detected by a hand-curated method (as represented by orange circles on the left top plot) with the rolonies suggested by the rolony densities (as indicated by red crosses on the left top plot). We see a broad agreement. Where there is a point of disagreement, we can visualize the signal intensities in all the voxels near that point. The two plots on the bottom-left show the original data from a spot that was detected by BarDensr, but not detected in the hand-curated results (as indicated as False Positive (FP) in the top left plot); the left columns show the original image and the right columns show the 'cleaned' image (similar to Fig 7, see Eq 2 for details). The red cross in each round indicates the channels that are activated by this barcode. These crosses line up well with the observed signal, suggesting BarDensr has correctly identified a new rolony. It appears that the hand-curated method failed to detect this rolony because of the presence of nearby rolonies, leading to a mixed signal; BarDensr is specifically designed to handle these kinds of confusing situations. The two plots on the bottom-right show a spot which is detected in the hand-curated result but not detected by BarDensr (as indicated as False Negative (FN) in the top left plot). We show both the original data and the cleaned data, as in the bottom left plots. In this case, the data do not appear to support the presence of a rolony, suggesting BarDensr correctly rejected this region as a rolony and the the hand-curated approach labeled it incorrectly. We conjecture that the hand-curated approach misidentified this as a spot because of the signal arising from a nearby rolony in round 7, channel 4; this again created a mixture of signals BarDensr was better equipped to recognize.
(TIF)

## Acknowledgments

We thank Abbas Rizvi, Li Yuan, Taiga Abe, Daniel Soudry, Ruoxi Sun, Darcy Peterka, and Ian Kinsella for many helpful discussions.

## Author Contributions

**Conceptualization:** Shuonan Chen, Jackson Loper, Xiaoyin Chen, Alex Vaughan, Anthony M. Zador, Liam Paninski.

**Data curation:** Shuonan Chen, Jackson Loper, Xiaoyin Chen.

**Formal analysis:** Shuonan Chen, Jackson Loper.

**Funding acquisition:** Xiaoyin Chen, Anthony M. Zador, Liam Paninski.

**Project administration:** Liam Paninski.

**Software:** Shuonan Chen, Jackson Loper.

**Supervision:** Anthony M. Zador, Liam Paninski.

**Writing – original draft:** Shuonan Chen, Jackson Loper, Liam Paninski.

**Writing – review & editing:** Shuonan Chen, Jackson Loper, Xiaoyin Chen, Liam Paninski.

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
