## [Decision Letter · Decision Letter 0]

20 Sep 2020

Dear Ms. Chen,

Thank you very much for submitting your manuscript "BARcode DEmixing through Non-negative Spatial Regression (BarDensr)" for consideration at PLOS Computational Biology. As with all papers reviewed by the journal, your manuscript was reviewed by members of the editorial board and by several independent reviewers. The reviewers appreciated the attention to an important topic. Based on the reviews, we are likely to accept this manuscript for publication, providing that you modify the manuscript according to the review recommendations.

All reviewers agreed that the paper is well written and describes a useful tool, but all made some comments on elements of clarity of the manuscript or the code notebooks. Please address these and consider the requests from R1 for additional results or text to address the potential limits and applications of the method.

Sincerely,

Emma Claire Robinson

Associate Editor

PLOS Computational Biology

Weixiong Zhang

Deputy Editor

PLOS Computational Biology

[LINK]

Reviewer's Responses to Questions

**Comments to the Authors:**

Reviewer #1: The manuscript by Chen et al, describes a new more efficient computational method to decode spots in images generated by combinatorial in situ labeling for multiplexed transcript detection. As experimentalist, and being a developer of chemistries of such in situ methods, I appreciate the development of better computational tools to increase the amount quality of information that can be extracted from these experiments. I have very limited training in mathematics, so I cannot judge whether the mathematical solutions make sense, but the general assumptions made about the nature of the data seem accurate. The paper is written in a very pedagogical way, so even a poorly theoretically trained experimental scientist like myself can follow the line of reasoning quite well. Some comments:

- In figure 4 and specially Figure 5, when assessing the performance of BarDensr compared to other methods, in all the situations BarDensr performs well. One would maybe like to see, especially in figure 5, how far can one go and when the method starts to fail, by adding more extreme cases with even more dense spots in the simulated data.

- When discussing how computationally expensive this method is, it would be good to do that in comparison with the other methods used to benchmark the method.

- One of the main problems when analyzing image-based spatially resolved RNA sequencing methods are stitching problems and misalignments between cycles. It would be good to include a paragraph about this.

- Here and there, I got lost when navigating between main figures and supplementary ones. I would prefer is more of the illustrations found in the supplementary figures could be added to the main figures.

- As a molecular biologist, I would have liked to see the method applied to larger piece of a biological specimen, like a region of the mouse brain displaying a high diversity of cell types, just to see what improvements can be achieved in terms of number of transcripts detected per cell, and whether the distribution of cell-type specific markers become tighter (and less dispersed), the letter being an indicator of improved specificity in barcode identification.

Reviewer #2: Chen et al., provide a reconstruction approach to spatial transcriptomics imaging data to identify underlying rolony densities. Given the application to cutting-edge spatial technologies I believe this method may be a useful tool for researchers analyzing spatial transcriptomics data. Please consider the following suggestions that may improve the present work.

1. The use of "spatial regression" in the title should be removed as it may be confusing for readers who may think this approach uses spatial models whereas the current paper uses non-negative least squares.

2. The authors used a partial subtraction approach to assess barcode-specific contribution. However, the overall reconstruction error should be reported using the flattened vector containing the observed pixel values.

3. Since the current framework uses some tensorflow functions under the hood, I am wondering how this approach would compare to ANNs and CNNs? Have the authors compared the present methodologies to neural network approaches based solely on reconstruction error as well as computation times?

4. The google collab notebook fails when generating the rez object (rez=bardensir.singlefov.process()) - "using a `tf.Tensor` as a Python `bool` is not allowed:"

Minor:

1. Are the 28 Figures supposed to be part of the main manuscript? Is there a reason for not labelling some as supplemental figures (e.g. Supplementary Figure 1 etc)?

Reviewer #3: Review PCOMPBIOL-D-20-01434 BARcode DEmixing through Non-negative Spatial Regression (BarDensr)

5/Sep/2020

Review:

First, I think this a very good methods paper.

In this submission, the authors present BarDensr, an open-source package that uses methods based on physical imaging parameters to demix rolony signals, a necessary step where convolution of signals may arise from insufficient technical resolution and other confounding factors.

The submission as it reads provides several steps of a necessarily complex model that can branch into different applications. The data analysis process as I understand it is, 1. User does data preparation (appendix A); 2. BarDensr employs the sparse non-negative regression framework to estimate parameters (F, rho, alpha, b; a; phi) and learns the model with a constrained optimization problem (this is where the user-interactive estimation of the constraining omega can occur as well); 3. BarDensr uses a blob-finding algorithm to find the rolonies for each barcode in the per-barcode images and thus deliver the location and the amount of gene expression in each rolony for a particular barcode representing the gene product.

The methods are well reasoned, and the Method section's descriptions are readable, though somewhat lacking in procedural flow that would help a reader understand the stepwise analysis used for a typical workflow in a practical application. The appendices and figures were instructive. The supporting links such as GitHub and the Python notebook and all the associated support files will have value to the user community, although one of the NeuroCAAS links was dead (BarDensr Bash Script Link). There is no tutorial I can find on how to use the package outside the bardensr_example.ipynb file, which was useful as an example, but it's unclear how this particular external resource will help a user correctly use BarDensr in a typical workflow as it is not well annotated nor explains the observations in depth. I will not dwell on those resources (though they are cited, and ultimately you will want to have a well annotated tutorial notebook than what is provided in the current ipython resource in order to have your method rapidly adopted.)

My questions/requests:

1. While I am not in favor of over-simplifying methods descriptions, I feel it would benefit your user community to briefly outline the steps utilized in a typical BarDensr workflow very specifically from the perspective of the user. This should include when to employ additional features in the workflow, such as interactive estimation of omega or estimation of good fit.

2. Since data preprocessing will probably be an important consideration in a typical analysis: Are there particular data preprocessing approaches that are optimal for satisfactory BarDensr analysis? How would a user estimate performance, perhaps with the SVD approach shown in Fig 10? Is there a suggested way to iterate to find the best set of parameters for data preprocessing?

**Have all data underlying the figures and results presented in the manuscript been provided?**

Reviewer #1: Yes

Reviewer #2: Yes

Reviewer #3: Yes

PLOS authors have the option to publish the peer review history of their article (what does this mean?). If published, this will include your full peer review and any attached files.

Reviewer #1: No

Reviewer #2: No

Reviewer #3: No
---

## [Decision Letter · Decision Letter 1]

27 Jan 2021

Dear Ms. Chen,

Thank you very much for submitting your manuscript "BARcode DEmixing through Non-negative Spatial Regression (BarDensr)" for consideration at PLOS Computational Biology. As with all papers reviewed by the journal, your manuscript was reviewed by members of the editorial board and by several independent reviewers. The reviewers all agree that the paper is ready for publication with exception of a recommendation that figures are reorganised and merged for improved readability (see R1). We therefore propose a minor revision.

Sincerely,

Emma Claire Robinson

Associate Editor

PLOS Computational Biology

Weixiong Zhang

Deputy Editor

PLOS Computational Biology

[LINK]

Reviewer's Responses to Questions

**Comments to the Authors:**

Reviewer #1: The authors have adequately responded to the comments and questions. One of the points of criticism is the arrangement of the figures. There are 12 main figures, some with very little content and then 18 supplemental figures, whereof many are really repetitive in nature. To increase the readability, I would suggest that some figures are merged (eg. 1+2+3; 4+5; and 7+8+9). Many of the repetitive elements of the supplementary figures could be reduced in order also to merge some of them.

Reviewer #3: Thank you for the changes. In the response, you wrote that your method is not a pipeline. I meant as your "user base" being actual bioinformaticians who may want to tweak paramters. But I understand what you're saying about dropping it into a pipeline.

**Have all data underlying the figures and results presented in the manuscript been provided?**

Reviewer #1: Yes

Reviewer #3: Yes

PLOS authors have the option to publish the peer review history of their article (what does this mean?). If published, this will include your full peer review and any attached files.

Reviewer #1: No

Reviewer #3: No
---

## [Editor Report · Decision Letter 2]

13 Feb 2021

Dear Ms. Chen,

We are pleased to inform you that your manuscript 'BARcode DEmixing through Non-negative Spatial Regression (BarDensr)' has been provisionally accepted for publication in PLOS Computational Biology.

Best regards,

Emma Claire Robinson

Associate Editor

PLOS Computational Biology

Weixiong Zhang

Deputy Editor

PLOS Computational Biology

---

## [Editor Report · Acceptance letter]

4 Mar 2021

PCOMPBIOL-D-20-01434R2 

BARcode DEmixing through Non-negative Spatial Regression (BarDensr)

Dear Dr Chen,

I am pleased to inform you that your manuscript has been formally accepted for publication in PLOS Computational Biology. Your manuscript is now with our production department and you will be notified of the publication date in due course.

With kind regards,

Andrea Szabo
